# Phenotypic and Quality Traits of Chickpea Genotypes under Rainfed Conditions in South Italy

Mohamed Houssemeddine Sellami [1,*], Antonella Lavini [1,*] and Cataldo Pulvento [2]

1    National Research Council of Italy (CNR), Institute for Agricultural and Forestry Systems in the Mediterranean (ISAFOM), 80055 Portici, NA, Italy

2    Department of Agricultural and Environmental Science, University of Bari "A. Moro" Via Amendola, 165/A, 70126 Bari, BA, Italy; cataldo.pulvento@uniba.it

\*    Correspondence: mohamed.sellami@isafom.cnr.it (M.H.S.); antonella.lavini@cnr.it (A.L.)

**Abstract:** Chickpea (*Cicer arietinum* L.) is an important cool-season food legume crop that is mainly cultivated as a rainfed crop. This study was conducted in Italy between 2017 and 2019 to evaluate the stability of seed yield (SY), biomass (AGB) and 1000 seed weight (THS), and to assess the seed quality of 12 kabuli chickpea accessions under field conditions. The likelihood-ratio test revealed significant effects of genotype only for the SY and THS. The environment and genotype × environment interaction (GEI) effects were highly significant for all variables. We found that the environment (year) and GEI explain 55.72% and 20.87% of the total seed yield variation, respectively. Most chickpea accessions showed sensitivity to frost conditions in the third growing season. No relationship was observed between the yield and the protein content in Kabuli chickpea. Among the accessions, Ares and Reale showed the best performance under all environmental conditions, and the Reale was the most stable chickpea.

**Keywords:** *Cicer arietinum*; genotype × environment interaction; stability analysis; seed quality

## 1. Introduction

Chickpea (*Cicer arietinum* L.) is an annual plant that belongs to the leguminous family, comprising a variety of beans, grass pea, soybeans, and lentils. Chickpea is native to the Mediterranean region and the Middle East [1]. It is a legume widely grown for its nutritious seeds. The seeds are high in fiber and protein and are a good source of iron, phosphorus, and folic acid [2]. In Italy, the cultivation of this legume in the last 20 years has progressively increased from over 4000 hectares to more than 20,000 hectares [3,4]. In 2020, approximately 33.5 thousand tonnes of chickpeas were produced in Italy on approximately 19,000 hectares, yielding an average of approximately 1.76 tonnes per hectare [4].

There are two distinct types of chickpea, called Desi and Kabuli chickpea, differing in terms of seed morphological traits. The Kabuli type with large, smoother, light-colored seeds, is grown in temperate regions (Mediterranean countries), while the Desi type is grown in semi-arid tropics (India), usually with yellow to black seed color, smaller in size, and a rough surface [5,6].

The chickpea yield production, like other crops, is affected by a number of biotic and/or abiotic factors. Among abiotic restrictions, drought and cold (freezing and chilling temperatures) are the most important factors limiting chickpea production [7,8]. The long taproot of the chickpea plant allows it to use water from a greater depth than other pulse crops [9], for this reason, the chickpea is considered as a drought-tolerant crop. In Mediterranean regions, the chickpea is sowing in winter [10]. However, cold during the cropping season is considered an important problem for winter-sown chickpea in the Mediterranean regions [11]. As reported by Croser et al. [12], freezing stress occurs during the seedling and early vegetative stages of crop growth in these regions. In general, chickpea

seems to adapt well to different kinds of environments in Italy. The production response is, however, closely linked to the availability of water during the production phases [13].

In Italy, the improvement of the genetic activity of this legume species was not particularly intense and only received due attention after the mid-1980s through the evaluation of the genetic variability of Italian and foreign germplasm [14]. Production stability and grain quality are the main objectives of the genetic improvement of chickpea. Recently, a scientific genetic selection activity has also been developed for selecting chickpea varieties rich in protein content and suitable for mechanization [14]. In Italy, some local landraces that are handed down through generations and characteristic of the cultivable habitat are cultivated [15]. These ecotypes are characterized by excellent resistance to different environmental conditions but are low-yielding and heterogeneous with regards to morphological traits, such as grain size and, often, very sensitive to disease incidence and parasitic attacks [14]. Today there is a growing interest in grain legumes, representing an important alternative protein source to meat for the future [16–18].

A large number of statistical procedures have been developed to enhance the breeder's understanding of genotype × environment interaction (GEI) and the stability of genotypes [19,20]. The additive main effects and multiplicative model (AMMI) is a commonly used method of analysis for stability. It can allow the characterization of the environment according to more variables [21]. According to Bassi and Sanchez-Garcia [22], these approaches to the GEI analysis allow for the characterization of the genotypes as "widely adapted" or "specifically adapted" to one environment, group of environments, or specific environmental variables, such as climate. The AMMI biplot analysis is considered to be an effective tool for graphically diagnosing GEI patterns [23], but fails in some aspects, such as accommodating a linear mixed-effect model (LMM) when one factor is fixed and others are randomly structured [24]. For this reason, when genotypes or environments (or both) are considering random effects, the best linear unbiased prediction (BLUP) offers the potential to improve the predictive accuracy of those effects [25].

The objectives of this study were to (1) evaluate the SY, yield components, and quality parameters of 12 chickpea accessions over three consecutive years and (2) determine the stability of yield for each accession.

## 2. Materials and Methods

### 2.1. Experimental Site and Climatic Data

Three-year long screening trials on chickpea were carried out in Vitulazio (Caserta, Italy) at the experimental research station of CNR-ISAFoM (41°12′ N and 14°20′ E, 23 m above the sea level), during three growing seasons of the years 2017, 2018, and 2019. The climate is typically Mediterranean and sub-humid, characterized by the 43-year average annual rainfall of 897 mm, mostly concentrated in autumn and winter (October to March). The annual reference evapotranspiration (ET0) estimated by the Penman–Monteith equation, according to Allen et al. [26], in the region is an average of 1074 mm between 1976 and 2019.

The main weather parameters considered for this study, including solar radiation, air temperature, relative humidity, and precipitation, were obtained from a standard agrometeorological station (iMetos ag. mod. IMT 280. Pessl Instruments. AT), which is located about 30 m away from the experimental field. The soil has a clay-loam texture (clay, sand, and silt: 46.1%, 30.2%, and 33.7%, respectively) and is defined as Mollic Haplaquept [27]. The chemical and physical characteristics of the soil at the beginning of the experiment were as follows: pH 8.05, Kjeldahl total N 1.81 g kg$^{-1}$, organic C 9.1 g kg$^{-1}$, electrical conductivity ECe = 0.23 dS m$^{-1}$, and bulk density 1.28 kg dm$^{-3}$. The volumetric soil water content at field capacity was 0.37 m$^3$ m$^{-3}$, while the permanent wilting point was 0.13 m$^3$ m$^{-3}$.

### 2.2. Crop Management and Experimental Design

The plant materials used in the study are depicted in Table 1. The accessions examined in the present study were chosen as they are extensively cultivated in the marginal areas of Italy and for their high potential stability from a phenological point of view. The seeds for the trials were sown between 28 and 29 November, 7 and 10 December, and 26 and 30 October in the first, second, and third growing seasons, respectively. The harvest was done in June and July for all growing seasons. The sowing density of plots was 25 plants $m^{-2}$. The sowing was performed with the implementation of a randomized complete block design (RCBD) with three replicates. All accessions grown under field conditions. The plot size was 3 $m^2$, with a distance of 0.4 m between two rows and 0.10 m between two plants. Light soil cultivation at the end of summer, followed by arrowing just before the seeding, was carried out in order to prepare the seedbed.

**Table 1.** Accessions Tested in the Study and their Origins.

| Code | Accession Name | Purveyor | Country |
|------|----------------|----------|---------|
| G1 | Ares | Ingegnoli nursery | Italy |
| G2 | di Cicerale | IBBR-CNR | ″ |
| G3 | di Guardia dei Lombardi | ″ | ″ |
| G4 | di Spinazzola | ″ | ″ |
| G5 | Pascià | ISEA | ″ |
| G6 | Pl 572509 | Western Regional Plant Introduction Station | United States |
| G7 | Pl 572512 | ″ | ″ |
| G8 | Sultano | ISEA | Italy |
| G9 | W6 11361 | Western Regional Plant Introduction Station | United States |
| G10 | Reale | ISEA | Italy |
| G11 | Pl 572515 | Western Regional Plant Introduction Station | United States |
| G12 | Pl 572517 | ″ | ″ |

quotation marks (″) indicate when purveyor or country is repetitive.

### 2.3. Measurements

At maturity, a subplot of 1 $m^2$ in the middle of each plot was manually harvested and the total yield (SY), the 1000 seed weight (THS), and the above-ground biomass (AGB) were determined. The harvest index (HI) was calculated as the ratio between SY and AGB. Seed samples (200 g) from the subplots harvested in 2017 were chemically analyzed through hierarchical cluster analysis (HCA).

The crude protein content was measured using the Kjeldahl method (AOAC 920.152). In this study, 2 g of the sample was briefly subjected to digestion at 450 °C (PBI International mod. Mineral SIX) with 30 mL of $H_2SO_4$ with 96% concentration in the presence of 7 g of $K_2SO_4$ and 0.7 g of $CuSO_4$. The digested substance was alkalinized with 45% NaOH and then subjected to steam distillation with the help of a distiller (Buchi mod. B-324). The condensed distillate was gathered in an Erlenmeyer flask containing 25 mL of 0.25 N $H_2SO_4$. The sulp huric acid not neutralized by the ammonia present in the distillate was titled with 0.25N NaOH in the presence of an indicator containing methylene blue/methyl red mix. The ammonia rate was estimated based on the difference in terms of the sulphuric acid equivalents between those present before and after the gathering of the ammonia distillate. It was converted into protein by using 6.25 as the conversion factor [28].

The raw fat determination was carried out in accordance with the AOAC 922.06 method [29]. A Soxhlet extraction thimble was used to weigh 10 g of the sample. Then, 3 g of anhydrous $Na_2SO_4$ was added, and an absorbent cotton was used as a seal. Fat was extracted with hexane by using an automatic extractor (PBI International mod. Soxhraction). The hexane was first removed using a vacuum-packed distillation and then placed in a

stove for 1 h at 105 °C. The weight of the extracted fat was then compared to the initial 10 g of the sample.

The ash content in the ashes was also determined in accordance with the AOAC 900.02 method (International). About 10 g of the samples were weighed in a capsule that had been previously calibrated at 550 °C for 4 h and chilled in a silica gel dryer. Subsequently, the samples were burned on a low flame and incubated overnight in a muffle furnace (Heraeus mod. K1251F). Afterwards, the ashes were chilled in a silica gel dryer and weighed soon after they reached room temperature. The ash rate was determined as the ratio between the remnant mass and the original sample mass.

The starch content was determined in accordance with the Ewers polarimetric method (ISO 10520:1997(E)).

### 2.4. Statistical Analysis

The data collected during the three years of the experimental work were analyzed according to a randomized complete block design (RCBD). The combination of year and location was considered as an environment (in our study, we had three environments). Each dependent variable was preliminarily evaluated for normal distribution by the Shapiro–Wilk's test. In case the assumption of normality was violated, we transformed the data into the Box-Cox transformation [30].

Yield and yield components data were analyzed with a linear model using a mixed-model via REML/BLUP (restricted residual maximum likelihood/best linear unbiased prediction), according to Henderson [31]. On the other hand, the Weighted Average of Absolute Scores (WAAS) [24] was computed for quantifying the stability of 12 accessions conducted in three environments. The WAAS is computed considering all Interaction Principal Component Axis (IPCA) from the Singular Value Decomposition (SVD) of the matrix of genotype–environment interaction (GEI) effects generated by a linear mixed-effect mode. The multi-trait stability index (FAI-BLUP index) used to ranking of genotypes and genotypes selected. The FAI-BLUP is a multi-trait index based on factor analysis and ideotype-Design that was recently proposed by Rocha et al. [32]. It is based on factor analysis, where the factorial scores of each ideotype are designed according to the desirable and undesirable factors. Then, a spatial probability is estimated based on genotype-ideotype distance, enabling genotype ranking. The results allowed for conducting a single and easy selection process of accessions.

These approaches were carried out by using the software package metan-multi-environment trial analysis [33] in the R studio software [34]. Both environment and genotype were considered to be a random effect.

Yield and yield components were subjected to principal component analysis (PCA) to explore the relationships among the variables and the treatments and to determine the yield that traits were the most effective in discriminating between accessions. The PCA outputs included variable loading for each selected component and treatment component scores. Prior to PCA, the data was Box-Cox -transformed. The PCA was based on the Pearson correlation matrix. This analysis was carried out by using the software package FactoMineR [35] in the R studio software [34].

The Spearman correlation was used to determine the relationship yield, yield components, and seed quality parameters. This analysis was carried out with the help of the software package Corrplot [36] in the R studio software [34].

The HCA with a single linkage method that used Euclidean distances for the seed quality, such as fat, ashes, starch, protein content, and protein yield was carried out by using package clValid [37] in the R studio software [34]. All packages used in the statistical analysis are available through the Comprehensive R Archive Network (CRAN, https://cran.r-project.org (accessed on 1 March 2021)).

## 3. Results

### 3.1. Weather Conditions

The weather regimes during the three experimental seasons are shown in Figure 1 in terms of the maximum (Tmax) and minimum (Tmin) air temperature, solar radiation, and rainfall. The climatic trend was different from the serial data (1976–2019) reference trend, especially in the case of the volume and distribution of rainfall. The seasonal precipitation was 408, 782, and 819 mm in the first, second, and third growing seasons, respectively, of the experimentation period, while the historical average was 723 mm. After computing the deciles index (DI), which was designed by Gibbs and Maher [38] and is one of the most widely used drought indices, the second (2017–2018) and third (2018–2019) growing seasons were classified as normal (DI = 9 and 9, respectively), whereas the first (2016–2017) season was considered weak dry (DI = 4). The overall average Tmax during the first, second and third growing seasons were 20.80 °C, 19.19 °C, and 19.72 °C, respectively, greater than the 43-year average (18.92 °C). The trial carried out in the third season was characterized by a colder winter season (December to February) when compared to the two other growing seasons; 2019 growing season was 1.49 °C and 0.91 °C lower than the values recorded in the same period in both 2017 and 2018, respectively. During winter, the Tmin below 0 °C occurred for 10, 8, and 12 days in 2017, 2018, and 2019, respectively. Moreover, during flowering (March–April), minimum temperature below 0 °C occurred for 1 day in both 2017 and 2019. The overall average solar radiation during the first season was 2.21 and 1.80 (MJ/m$^2$/day) higher than the values recorded in the same period in both 2018 and 2019, respectively.

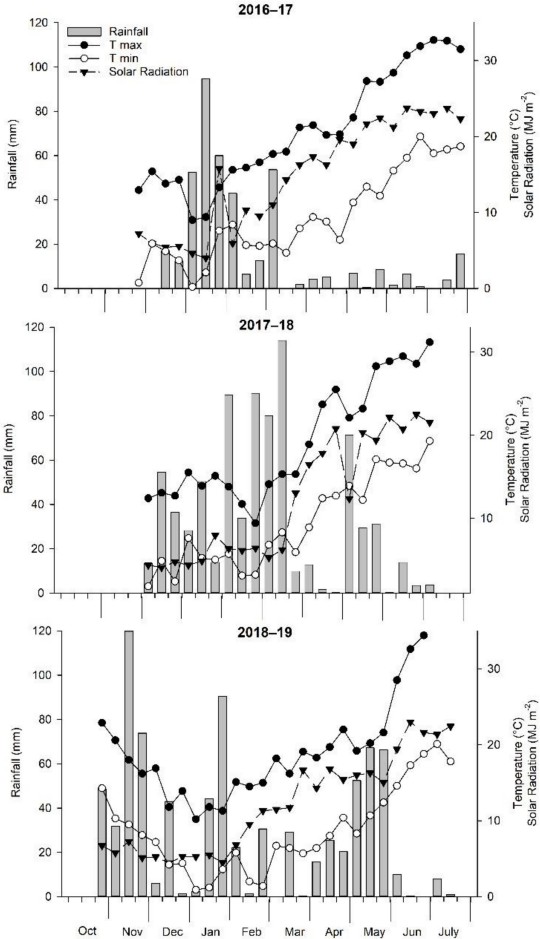

**Figure 1.** Rainfall, air temperature (Tmin and Tmax), and solar radiation distribution during the three experimental years.

### 3.2. Likelihood Ratio Tests and Variance Components

Table 2 shows the likelihood ratio test, estimated variance components, and SY mean of 12 chickpea accessions that were evaluated for three consecutive growing seasons. According to the likelihood ratio test, the genotype effect was significant only for the SY, AGB, and THS. The environment and GEI effects were highly significant for all variables. The SY, AGB, and THS differed markedly from year to year (Table 2), with the highest values of SY, AGB, and THS being recorded in the second season and the lowest values being recorded in the third (except for AGB). The effect of the year was also evident on the HI, where the highest value was recorded during 2017. The overall SY mean was 3.75 ± 1.67 t/ha. The GY was 37.59% and 64.74% less in the first and third seasons, respectively, compared to the second season, and the AGB was 48.21% and 11.01% less in the first and third seasons, respectively, compared to the second season. The largest proportion of variance in SY was explained with respect to the environment, with 55.72% being the total phenotypic variance (Figure 2). Amongst all traits, except for THS, the GEI variance ($\sigma ge^2$) was higher than genotypic ($\sigma g^2$) and residual variance ($\sigma r^2$) (Figure 2).

**Table 2.** Likelihood Ratio Test, Estimated Variance Components, and SY Mean of 12 Chickpea Accessions Evaluated for Three Consecutive Growing Seasons.

| Source of Variation | LRTe | LRTg | LRTge | Ve | Vg | Vge | $V_{E/B}$ | Vr | Growing Season | | |
|---|---|---|---|---|---|---|---|---|---|---|---|
| | | | | | | | | | 2017 | 2018 | 2019 |
| SY (t/ha) | 16.9 **** | 5.84 * | 73.9 **** | 3.07 | 0.974 | 1.15 | 0.0622 | 0.242 | 3.78 ± 1.38 [b] | 5.53 ± 1.90 [a] | 1.95 ± 1.16 [c] |
| AGB (t/ha) | 9.22 ** | 4.08 * | 96.7 **** | 17.3 | 12.5 | 20.3 | 0.332 | 2.83 | 8.98 ± 2.35 [c] | 17.34 ± 6.65 [a] | 15.43 ± 7.23 [b] |
| HI (%) | 25.4 **** | 0.00 ns | 47.5 **** | 215 | 0.0599 | 60.4 | 0.00 | 24.6 | 41.00 ± 7.85 [a] | 33.92 ± 11.54 [b] | 12.46 ± 7.06 [c] |
| THS (g) | 16.5 **** | 7.33 ** | 37.7 **** | 4470 | 3110 | 2740 | 0.00 | 1450 | 355.30 ± 63.91 [b] | 393.48 ± 84.09 [a] | 259.80 ± 98.67 [c] |

ns, *, **, **** indicate not significant difference, significant at $p \leq 0.05$, $p \leq 0.01$, and $p \leq 0.0001$, respectively. *LRTe*, *LRTg*, and *LRTge* are likelihood ratio tests for environment, genotype, and interaction g × e. *Ve*, *Vg*, *Vge*, $V_{E/B}$, and *Vr*, variance components for environment, genotype, interaction, block-within-environment, and residuals, respectively. SY = Seed yield; AGB = Above ground biomass; HI = Harvest index and THS = 1000 seed weight. Means followed by the same letter in each column are not significantly different according to the Tukey test ($p = 0.05$).

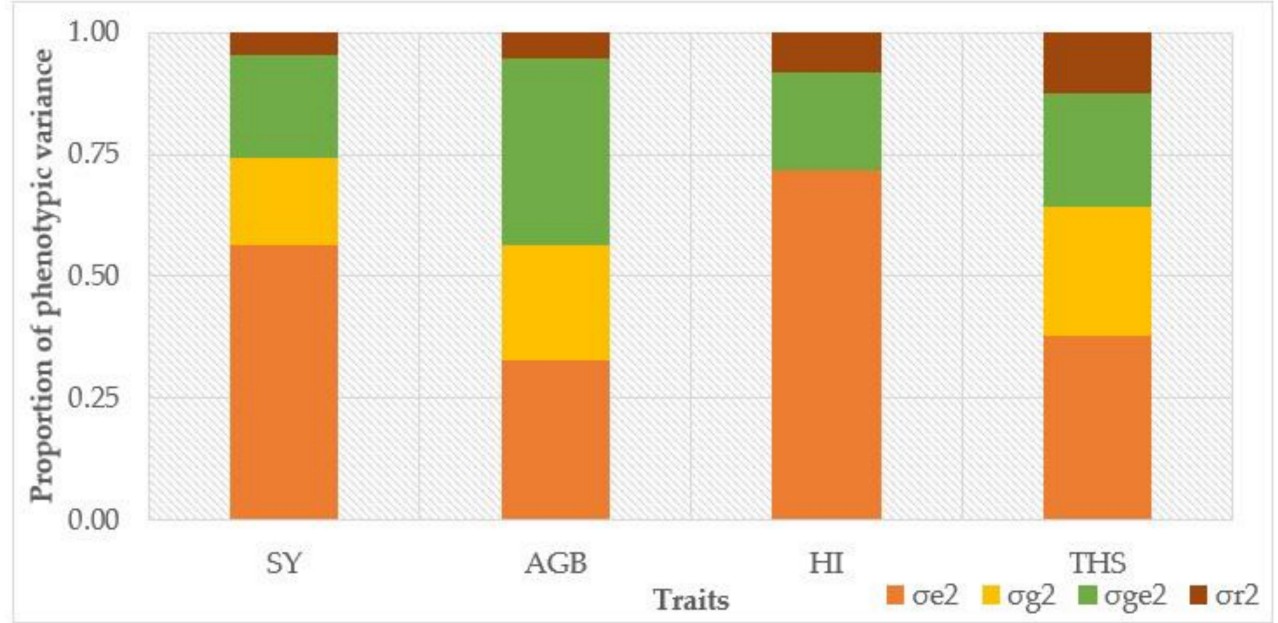

**Figure 2.** Proportion of the phenotypic variance for four chickpea traits (SY, AGB, HI, and THS) evaluated during three consecutive growing seasons. SY = Seed yield; AGB = Above ground biomass; HI = Harvest index and THS = 1000 seed weight. σe2 = environment variance; σg2 = genotype variance; σge2 = interaction g × e variance and σr2 = residual variance.

### 3.3. Principal Component Analysis and Seed Yield Response Factor

To obtain a comprehensive overview of the yield and yield components of chickpea in response to the accessions and growing seasons, the whole data set, including the climatic parameters during the three consecutive growing seasons, was subjected to PCA. The first two principal components (PCs) corresponded to eigenvalues higher than one and explained 84.5% of the cumulative variance for yield and yield components. PC1 (first component) accounted for 52.46%, while PC2 (second component) accounted for 32.04% of the cumulative variance for the yield and yield components (Figure 3). For chickpea, PC1 was positively and strongly correlated (>0.6) with increased yield, THS, and HI. PC2 was positively correlated with increased AGB (Figure 3).

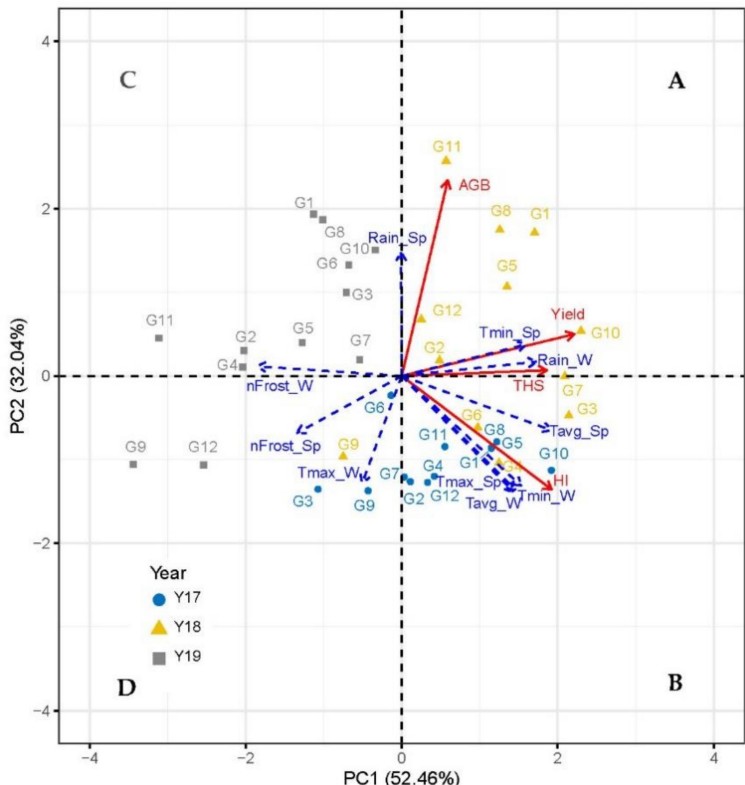

**Figure 3.** Principal component loading plots and scores of the PC1 and PC2 after PCA on AGB, SY, HI, and THS as a function of growing seasons and accessions. Box-Cox-transformed data was used for the PCA. The square indicates a Pearson correlation coefficient of 1. The red arrows represent the principal variables. Dotted blue arrows represent the supplementary variables (weather parameters). The values from 1 to 12 represent the different accessions (see Table 1 for more details). Tavg_W (average temperature in winter), Tmax_W (maximum temperature in winter), Tmin_W (minimum temperature in winter), Rain_W (rainfall in winter), nFrost_W (number of days per month where the minimum temperature dropped below 0 °C in winter), Tavg_Sp (average temperature in spring), Tmax_Sp (maximum temperature in spring), Tmin_Sp (minimum temperature in spring), Rain_Sp (rainfall in spring), nFrost_Sp (number of days per month where the minimum temperature dropped below 0 °C in spring). A, B, C, D indicate upper-right, lower-right, upper-left, lower-left quadrant, respectively.

In the current study, the positive side of PC1 (Figure 3), in particular the upper right quadrant (A), included most of the accessions for the 2018 growing season, where G10 (Reale), G3 (di Guardia dei Lombardi), and G7 (Pl 572512) were the most productive accessions. The treatments from the upper-left quadrant (C) with the most treatments coming from the 2019 growing season were characterized by lower growth and productivity (onlyhigh AGB was recorded during this season). The low crop growth and productivity

in 2019 were mainly related to the frost in winter and spring. The G9 (W6 11361), G11 (Pl 572515), and G12 (Pl 572517) were the lowest productive chickpea accession (Figure 3).

According to Figure 3, the variation of yield due to environment and genotype factors was quite large. with the lowest value of 0.01 t ha$^{-1}$ being generated for the accession G12 (Pl 572517) in 2018 growing season and the highest value of 8.63 t ha$^{-1}$ for the accession G1 (Ares).

Figure 4 shows chickpea yield variation among accessions across three cropping seasons. It can be seen from Figure 4 that the highest overall SY was recorded for the accession G1 (Ares) with 5.09 t ha$^{-1}$ and the lowest for the accession G9 (W6 11361) at 1.70 t ha$^{-1}$.

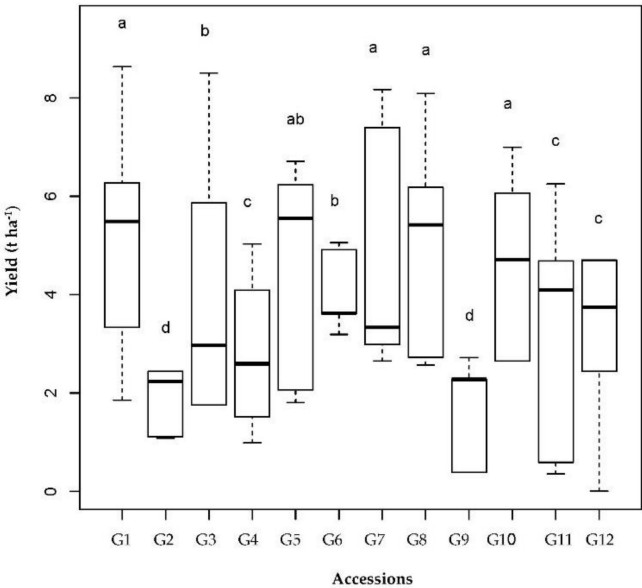

**Figure 4.** Boxplots of SY (t ha$^{-1}$) of the 12 chickpea accessions. Accessions followed by a different letter in each row are significantly different according to the Tukey HSD (honestly significant difference) test ($p = 0.05$). The values from 1 to 12 represent the different accessions (see Table 1 for more details). Box edges represent the upper and lower quantile with the median value shown in the middle of the box.

### 3.4. Yield Stability

Figure 5 shows the mean SY (t ha$^{-1}$) plotted against the coefficient of variation (CV) as a percentage for the 12 accessions. The CV varied from 18.11% (Pl 572509) to 67.84% (Pl 572515). It can be seen from Figure 5 that six accessions (Ares, Pascià, Pl 572509, Pl 572512, Reale, and Sultano) have high yield with low CV. The accession Ares and Sultano showed significantly larger yield means than W6 11361, the lowest yielding accession.

Figure 6 represents the biplot of the SY versus the weighted average of absolute scores for the best linear unbiased predictions of the GEI (WAASB) of the 12 chickpea accessions evaluated during three consecutive growing seasons. The genotypes or environments included in quadrant I can be considered unstable with high discrimination ability and with productivity below the grand mean. The growing seasons of 2019 (E3) and chickpea accessions G2(di Cicerale) and G9 (W6 11361) were included in this quadrant. In quadrant II unstable genotypes are included although with productivity above the grand mean. The environments included in this quadrant provide high magnitudes of the response variable (SY) and present a good discrimination ability. The growing season 2017 and 2018 (E1 and E2, respectively) and the accessions G3 (di Guardia dei Lombardi) and G7(Pl 572512) were included in this quadrant. Genotypes within quadrant III have low productivity but can be considered stable due to their lower values of WAASB. The lower this value, the more stable can the genotype be considered. The environments included in this quadrant can be considered as poorly productive and having low discrimination ability. The chickpea

accessions G4 (di Spinazzola), G11 (Pl 572515), and G12 (Pl 572517) were included in this quadrant. The genotypes within quadrant IV are considered to be highly productive and can be broadly adopted due to the high magnitude of the response variable (SY) and their high stability performance (lower values of WAASB). G1 (Ares), G5 (Pascià), G6 (Pl 572509), G8(Sultano), and G10 (Reale) were included in this quadrant. G10 (Reale) was the most stable chickpea accession because it had a smaller WAASB value.

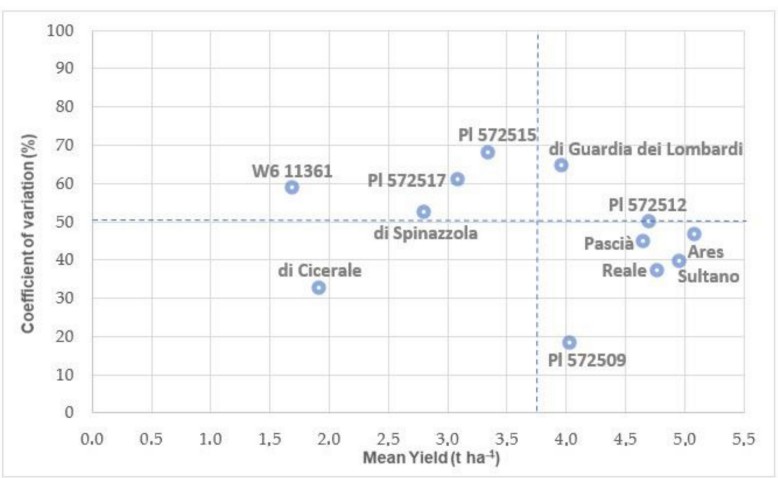

**Figure 5.** Mean SY (t ha$^{-1}$) plotted against the CV as a percentage for the 12 chickpea accessions. Dashed lines represent the average yield (vertical) and the CV across all the accessions (horizontal).

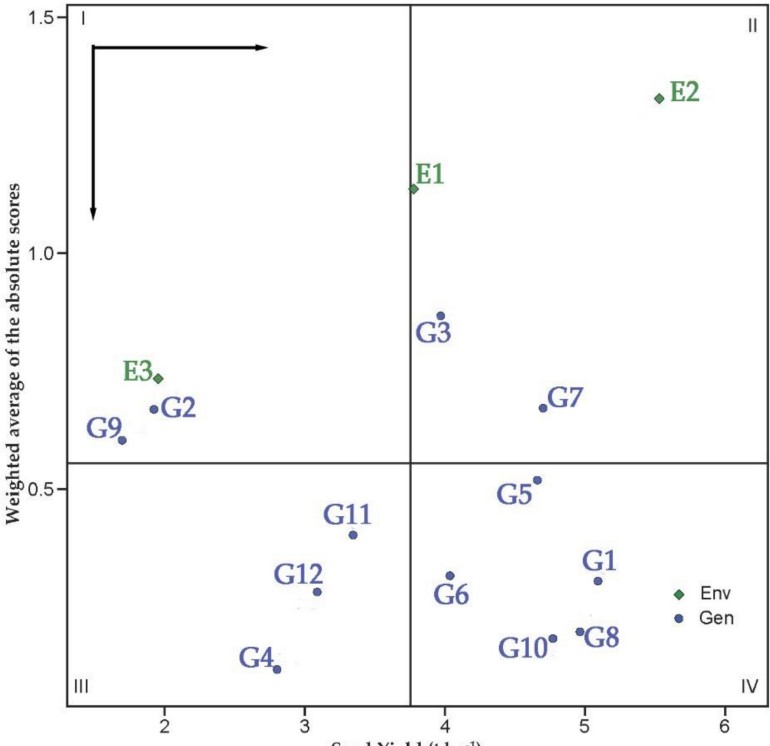

**Figure 6.** Biplot of the SY vs. the weighted average of absolute scores for the best linear unbiased predictions of the genotype × environment interaction (WAASB) of the 12 chickpea accessions evaluated during three consecutive growing seasons. Horizontal and vertical black arrows indicate the direction of the increase in yielding and stability, respectively. Values from 1 to 12 represent the different accessions (see Table 1 for more details). E1, E2, and E3 indicate: 2017, 2018, and 2019 growing season, respectively.

Figure 7 shows the ranking of the 12 chickpea accessions according to the multi-trait stability index (FAI-BLUP index). The first two accessions (G1 and G10) selected according to the FAI-BLUP index display balanced traits related to SY, AGB, HI, and THS, without assigning weights, free from multicollinearity and, therefore, should be selected as multi-trait high-performance accessions. The accession G8 was near to the selection intensity circle and could present interesting features.

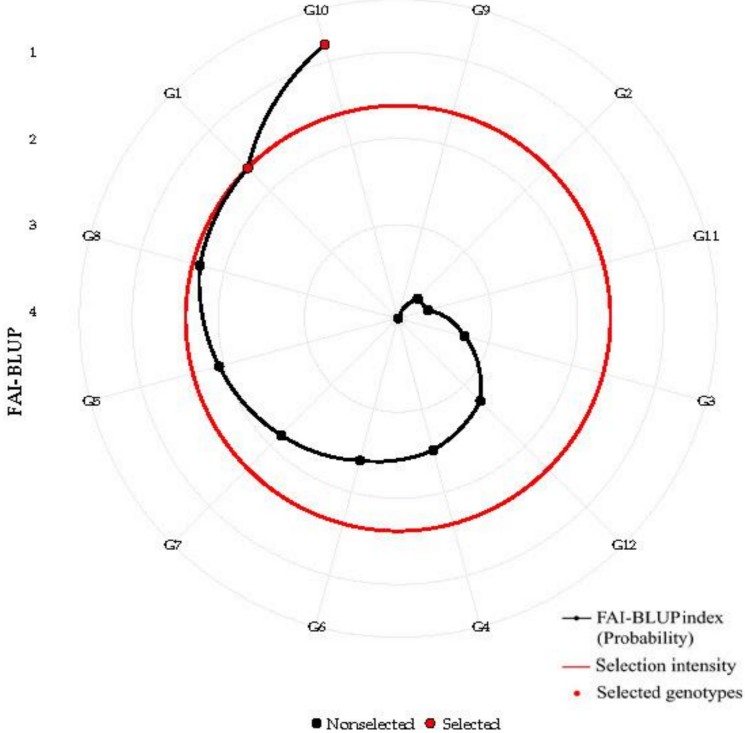

**Figure 7.** Genotype ranking and selected genotypes for the multi-trait stability index (FAI-BLUP index) considering a selection intensity of 15% (red circle). 1, 2, 3 and 4 indicate the multi-trait stability index.

*3.5. Yield Quality*

In the 2017 growing season, the seed quality analysis of the 12 chickpea accessions was conducted. The objective of yield quality analysis is to evaluate how the yield quality changes between the chickpea accessions under fixed cropping season. Figure 8 shows the dendrogram generated by HCA of dissimilarities among the chickpea accessions on their Euclidean distances for the seed quality, such as fat, ashes, starch, protein content, and protein yield. The HCA revealed five major chickpea groups (Figure 8).

The four Kabuli chickpeas G5, G6, G10, and G10 Pascià caused similar responses in the seed quality parameters and, therefore, were grouped into one cluster: Group 1. Group 2 included two kabuli chickpeas, G3 and G9. Group 3 clustered six Kabuli chickpeas together, including G1, G2, G4, G7, G8, and G12.

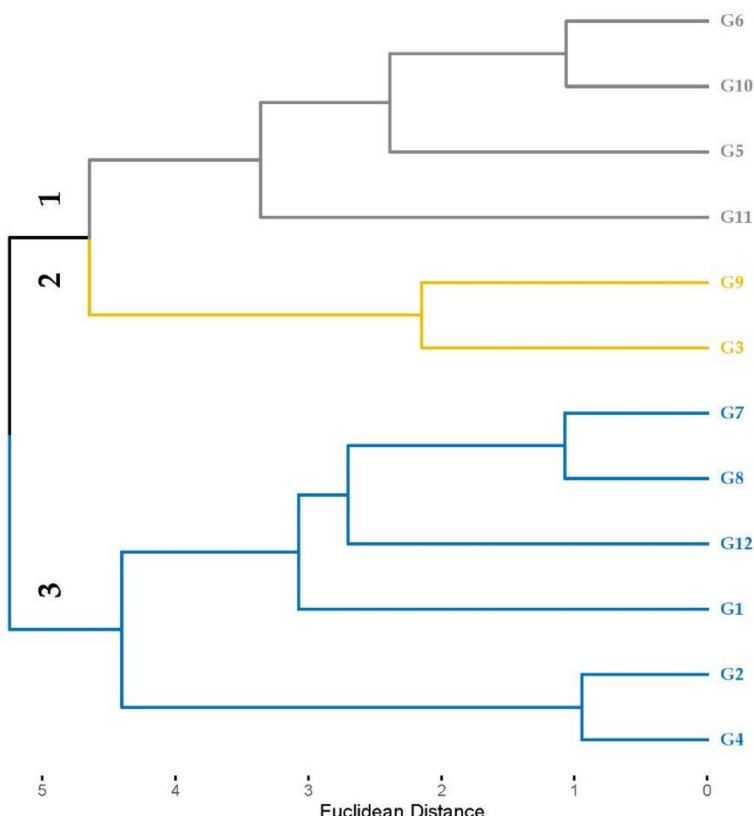

**Figure 8.** Dendrogram generated by the HCA of dissimilarities among the chickpea accessions on their Euclidean distances for the seed quality parameters, such as fat, ashes, starch, protein content, and protein yield. Each joining (fusion) of three clusters is represented on the graph by the splitting of a horizontal line into two horizontal lines. The horizontal position of the split, as indicated by the short vertical bar, gives the distance (dissimilarity) between two clusters. Three major accessions groups were identified by this analysis and their numbers have been presented above the differentiating node.

According to Table 3, Group 1 had a higher value for ashes, fat, and protein yield. Among the accessions in this group, G5 is the highest-performing chickpea accessions. On the contrary, Group 2 chickpea accessions had the lowest values for ashes, protein content, fat, and protein yield and the highest value of starch. Group 3 was characterized by the highest value of protein content.

**Table 3.** Mean Quality Parameters of Three Major Chickpea Groups during the 2017 Growing Season.

|  | Starch (g/100 g Dry Weight) | Ashes (%) | Protein Content (%) | Fat (%) | Protein Yield (kg ha$^{-1}$) |
|---|---|---|---|---|---|
| Group 1 | 43.08 ± 1.78 | 3.05 ± 0.06 | 19.75 ± 0.56 | 6.32 ± 0.48 | 828.15 ± 115.43 |
| Group 2 | 43.75 ± 0.35 | 2.86 ± 0.03 | 19.55 ± 0.49 | 1.75 ± 0 | 558 ± 328.27 |
| Group 3 | 42.1 ± 0.57 | 3.01 ± 0.15 | 21.17 ± 0.68 | 2.41 ± 1.41 | 793.47 ± 272.63 |

Values reported as Mean ± SD.

Table 4 shows the results of the Spearman correlation (*n* = 12) of the yield, yield components, and the seed quality for 12 Kabuli chickpea accessions during the 2017 growing season. According to Table 4, there is no significant correlation between quality parameters and yield and yield components—except for protein yield had a positive significant correlation with yield (r = 0.96, *p* ≤ 0.0001) and AGB (r = 0.77, *p* ≤0.001). A good correlation (r = 0.85) was also found between the yield and the AGB in the present study.

**Table 4.** Correlations Coefficients among Yield, Yield Components (THS, AGB, and HI), and Seed Quality (Protein content, Fat, Starch, and Ashes) Variables for 12 Kabuli Chickpea accessions during the 2017 Growing Season.

| Variables | Starch | Ashes | Protein Content | Fat | Protein Yield | Yield | THS | AGB | HI |
|---|---|---|---|---|---|---|---|---|---|
| Starch | **1** | | | | | | | | |
| Ashes | −0.03 | **1** | | | | | | | |
| Protein content | −0.06 | 0.42 | **1** | | | | | | |
| Fat | 0.19 | 0.14 | −0.27 | **1** | | | | | |
| Protein yield | −0.23 | −0.30 | 0.15 | 0.31 | **1** | | | | |
| Yield | −0.29 | −0.35 | −0.02 | 0.39 | **0.96** | **1** | | | |
| THS | −0.46 | 0.29 | 0.23 | −0.15 | 0.25 | 0.30 | **1** | | |
| AGB | −0.19 | −0.16 | −0.08 | 0.50 | **0.77** | **0.85** | 0.17 | **1** | |
| HI | 0.06 | −0.16 | 0.08 | 0.03 | 0.51 | 0.43 | 0.41 | 0.01 | **1** |

Values in bold are different from 0 with a significance level alpha = 0.05. AGB, above-ground biomass; HI, harvest index; THS, 1000-seed weight.

## 4. Discussion

Phenotypic information of the 12 kabuli chickpea accessions was collected during three consecutive growing seasons at the experimental research station of CNR-ISAFoM in South Italy. According to Ezatollah et al. [39], the analysis of genotype by environment data must start with the examination of the magnitude and nature of genotype through environmental interaction. In this study, the genotypic variance ($\sigma g^2$) explained only a small portion of the total phenotypic variance for all traits, indicating that the environments were more different from one another than the genotypes. Similar findings were reported by Khan et al. [40,41] and Tilahun et al. [42]. The likelihood ratio test shows the presence of a significant effect of GEI for all traits. A significant GEI indicated that genotypic response varied with the environment. The presence of this significance in chickpea was reported by various authors, such as Tilahun et al. [42], Bozoglu and Gulumser [43], and Mohammadi et al. [44].

The considerably high proportion of phenotypic variance in SY, which can be explained by environment variance ($\sigma e^2$) alone, can be attributed to a significant annual climate change due to a high number of frost days (<0 °C) in the winter and spring of 2019 compared to other growing seasons. The frost nights in the third season would certainly lead to a further reduction in yield. Similar findings were reported by Nezami et al. [8] and Croser et al. [12]. The second growing season (2018) had favorable environmental conditions for the crops when compared to the two other growing seasons.

In our experiment, we found that the genotypic variance ($\sigma g^2$) on THS is larger than the GEI variance ($\sigma g \times e$) and residual variance ($\sigma r^2$). This shows that the variation for THS existed among the accessions, and it calls for a specific form of selection of accessions according to these traits in future breeding programs.

Among the 12 chickpea accessions, the chickpea accessions G1 (Ares), G5 (Pascià), G6 (Pl 572509), G8 (Sultano), and G10 (Reale) performed fairly well in all the environments, as assessed in this study, which demonstrated their ability to produce reasonable yields under very diverse climatic conditions. G10 (Reale) was the most stable chickpea accession because it had smaller BLUP-based stability index (WAASB) values.

The multi-trait index based on factor analysis and ideotype-Design (FAI-BLUP index) indicated the two chickpea accessions G1 (Ares) and G10 (Reale) showed the highest performance and predicted balanced and desirable genetic gains for all the considered traits. The accessions selected by the FAI-BLUP index have the potential to improve all the traits simultaneously. According to Olivoto et al. [45], the FAI-BLUP index provides a unique selection process and considers the correlation structure among the traits.

Concerning the seed quality traits, the total starch was negatively correlated with THS ($r = -0.46$, ns). This is contradictory to the results reported previously [46,47]. According to Frimpong et al. [47], the relationship between starch concentration and seed weight was positive in both Kabuli ($r = 0.16$, $p = 0.05$) and Desi ($r = 0.11$, ns) varieties but was not significant in the latter.

Throughout our study, the protein content ranged from 19.20% to 21.70% (data not shown). Zaccardelli et al. [48] reported very similar results. The protein content values were lower for chickpea than the other leguminous species, such as grass peas, with values up to 24% [49]. In our study, the absence of the correlation between SY and protein content in Kabuli chickpea was demonstrated. Frimpong et al. [47] found the same result for the Kabuli variety (r = −0.06, ns). The relationship between SY and protein content for Kabuli chickpea is similar to that reported in the case of fava beans and grass pea by several studies [49–51]. The absence of correlation in the Kabuli chickpea accession could probably be attributed to the capacity of some chickpea accession, due to the fixation of biological nitrogen.

Finally, the seed quality of W6 11361 and di Guardia dei Lombardi were very poor (except of starch parameter) and significantly different from all the other tested accessions. This agreed with the agronomic result.

## 5. Conclusions

In the present study, 12 Kabuli chickpea accessions were evaluated for three different environmental and some stability parameters using CV and the weighted average of absolute scores for the best linear unbiased predictions (WAASB) of the GEI in SY, AGB, HI, and THS. The results show that the proportion of variance explained by the environmental factor (years) was much more important than other factors (genotype and GEI), at least in the case of SY, THS, and HI, whereas the effect of GEI was more significant on the AGB. The 12 chickpea accessions only showed sensitivity to frost conditions in the third growing season, during which some night frost events occurred.

According to the results of coefficient of variation and weighted average of absolute scores for the best linear unbiased predictions of the GEI (WAASB), Ares, and Reale were determined to be multi-trait high-performance accessions in all environments. Reale was the most stable chickpea accession because it had a smaller WAASB value and the accession Sultano could present interesting features.

Based on the results of the qualitative analysis, six Kabuli chickpeas (Pl 572512, Pl 572517, Sultano, Ares, di Cicerale, and di Spinazzola) showed high percentages of total protein, while two chickpeas (W6 11361 and di Guardia dei Lombardi) showed a high percentage of starch.

The evaluation of three selection indices, i.e., yield stability, yield potential, and yield quality, will help choose the best chickpea accessions for the marginal areas in South Italy. The FAI-BLUP index acts as useful tools for breeders and agronomists who aim to select accessions according to their mean performance and stability with a consideration of a number of traits.

**Author Contributions:** Conceptualization, A.L. and C.P.; Methodology, A.L. and C.P.; Formal analysis, M.H.S.; Investigation, A.L. and C.P.; Resources, A.L. and C.P.; Data curation, M.H.S., A.L. and C.P.; Writing—original draft preparation, M.H.S.; Writing—review and editing, M.H.S., C.P. and A.L.; Visualization, M.H.S.; Supervision, C.P. and A.L. All authors have read and agreed to the published version of the manuscript.

**Funding:** This research was funded by European Union's Horizon 2020 research and innovation program (Protein2Food project), grant number 635727.

**Institutional Review Board Statement:** Not applicable.

**Informed Consent Statement:** Not applicable.

**Data Availability Statement:** The datasets and the R codes used in this study are available from the authors upon reasonable request.

**Acknowledgments:** The authors thank Calandrelli Davide for technical support to field trial, in additions, the authors thank Angela Rosa Piergiovanni from IBBR-CNR and the. Western Regional Plant Introduction Station for genetic material used.

**Conflicts of Interest:** The authors declare no conflict of interest.

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
