# Peer review of "Phenotypic and Quality Traits of Chickpea Genotypes under Rainfed Conditions in South Italy"

_agronomy, doi:10.3390/agronomy11050962_

Round 1

Reviewer 1 Report

The study, aimed to evaluate the seed yield, yield components
and quality parameters of 12 chickpea accessions over three consecutive years and to determine the stability of yield for each accession is interesting, of high presentation quality and scientific soundness. 

Author Response

We are grateful to the Reviewer for the very useful comments.

Reviewer 2 Report

This study provides results about agronomic and quality traits evaluated in different Italian chickpea varieties together with four accession from USDA. Yield components were evaluated in three different seasons following an appropriate experimental design. Also, quality seeds components were included in the analysis together with detailed climatic data in different seasons. Authors performed a deep statistical analysis. In general, the manuscript is well written, and information could be interesting for chickpea breeder’s community. However, authors indicate in discussion that the genotypic variance (σg2 ) explained only a small portion of the total phenotypic variance for all traits” . This study analyzes the behavior of different genotypes so, maybe, experimental design should be revied. I should recommend minimizing environmental effect for yield components analyzing data in central rows of experimental plot.

I also include some questions or suggestions to be reviewed:

Plant material: Please explain in the text why you choose those accessions from USDA, any relevant agronomic trait?

Line 112: What is here the meaning of “an elementary plot”? every elementary plot (3 repetition per year) or a single plot per year? It is not clear. Besides, yield is usually evaluated in central rows of elementary plot to avoid the effect of the borders, it seems that here this possibility was not considered.

Line 114: seeds harvested in 2017: that means that quality components were analysed in a single year, please explain in more detail the way of sampling per genotype for chemistry analysis, it is not clear.  

Table 1: All accessions are kabuli type, so just could be indicated in the text, it is not necessary to include a column in this table. Besides, to simplify the text of the table use quotation marks (“) when purveyor or country is repetitive

Table 2: could be removed. This information could be given in the text. Why those differences in sowing data in the same year? Please explain this situation, it is not usual to have differences in sowing date in the same field trial.

Author Response

Dear Reviewer,

Thank you indeed for reviewing the manuscript. I have uploaded my answers to your valuable comments. My coauthors and I hope the manuscript amelioration is sufficient for acceptance.

Kind regards.

Mohamed houssemeddine SALLAMI

Reviewer 3 Report

            This manuscript presents a substantial dataset and the statistical analysis is suitable.  Accordingly, my comments are few and minor.

            Tables 1 and 2.  Since the seed type in Table 1 is always the same, this column should be deleted.  The text has already made it clear that Kabuli chickpeas are the topic here.

            Table 2.  It would be better to list the accessions in Table 2 in the same order as in Table 1.

            Line 166 says that both environment and genotype were deemed random effects, but gives no explanation or reason.  Please explain why that choice is made.  Since the text focuses interest on particular genotypes, rather than Kabuli chickpeas in general, it would seem that genotypes are being discussed as if they were fixed effects.

            Figure 3 is spectacularly informative and makes a lot of sense.  In particular, winter rain is correlated with yield (but interestingly not spring rain), and number of frost days is negatively correlated with yield.  Because PCA can be done with different transformations of the data, the transformation used here should be stated in the text or figure legend.  Were the data standardized?  If so, by the traits/vectors, or by the points/genotype-years?

            Sometimes the environments are identified as E1, E2, and E3 and sometimes as Y17, Y18, and Y19.  Perhaps I missed something, but it was unclear to me which year corresponds to E1 etc.  Maybe always using Y17, Y18, and Y19 would be better because these labels have a direct and obvious meaning.

            Because this chickpea experiment was repeated over years but not across locations, perhaps a bit more could be said about how this research location relates to the growing region for chickpeas in Italy – and beyond Italy.  Are these findings most relevant for “marginal areas in South Italy” (line 435) in particular, or are they reliable for a larger region?

Author Response

(The authors gave the same response as above.)

Round 2

Reviewer 2 Report

I consider that the corrected version of the manuscript improved previous version. Authors have been clarified relevant aspects in methodology.